# Colistin Heteroresistance among Extended Spectrum β-lactamases-Producing *Klebsiella pneumoniae*

**DOI:** 10.3390/microorganisms8091279

**Published:** 2020-08-21

**Authors:** Felipe Morales-León, Celia A. Lima, Gerardo González-Rocha, Andrés Opazo-Capurro, Helia Bello-Toledo

**Affiliations:** 1Facultad de Farmacia, Departamento de Farmacia, Universidad de Concepción, Concepción 4030000, Chile; felimora@udec.cl; 2Laboratorio de Investigación en Agentes Antibacterianos (LIAA), Departamento de Microbiología, Facultad de Ciencias Biológicas, Universidad de Concepción, Concepción 4070386, Chile; cfernades@udec.cl (C.A.L.); ggonzal@udec.cl (G.G.-R.); 3Millennium Nucleus for Collaborative Research on Bacterial Resistance, MICROB-R, Concepción 4070386, Chile; 4Departamento de Prevención y Salud Pública, Facultad de Odontología, Universidad de Concepción, Concepción 4070386, Chile; 5Departamento de Medicina Interna, Facultad de Medicina, Universidad de Concepción, Concepción 4070386, Chile

**Keywords:** Colistin heteroresistance, *Klebsiella pneumoniae*, extended-spectrum β-lactamases

## Abstract

Colistin-heteroresistant (CST-HR) Enterobacterales isolates have been identified recently, challenging the clinical laboratories since routine susceptibility tests fail to detect this phenotype. In this work we describe the first CST-HR phenotype in extended-spectrum β-lactamase (ESBL)-producing *Klebsiella pneumoniae* isolates in South America. Additionally, we determine the genomic mechanisms of colistin heteroresistance in these strains. The CST-HR phenotype was analyzed by the population analysis profile (PAP) method, and mutations associated with this phenotype were determined by whole-genome sequencing (WGS) and the local BLAST+ DB tool. As a result, 8/60 isolates were classified as CST-HR according to the PAP method. From WGS, we determined that the CST-HR isolates belong to three different Sequence Types (STs) and four K-*loci*: ST11 (KL15 and KL81), ST25 (KL2), and ST1161 (KL19). We identified diverse mutations in the two-component regulatory systems PmrAB and PhoPQ, as well as a disruption of the *mgrB* global regulator mediated by IS1-like and IS-5-like elements, which could confer resistance to CST in CST-HR and ESBL-producing isolates. These are the first descriptions in Chile of CST-HR in ESBL-producing *K. pneumoniae* isolates. The emergence of these isolates could have a major impact on the effectiveness of colistin as a “last resort” against these isolates, thus jeopardizing current antibiotic alternatives; therefore, it is important to consider the epidemiology of the CST-HR phenotype.

## 1. Introduction

In recent years, the prevalence of multidrug-resistant (MDR) Enterobacterale isolates in clinical settings has been increasing alarmingly. Among this group, extended-spectrum β-lactamase (ESBL)-producers and carbapenem-resistant *Klebsiella pneumoniae* represent a serious threat to public health [1,2]. As such, polymyxin antibiotics, such as colistin (CST), are considered to be a “last-line therapy” to treat infections caused by highly resistant *K. pneumoniae* due to their microbiological activity [3] and pharmacodynamic properties [1,4]. Worryingly, colistin-heteroresistant (CST-HR) Enterobacterales isolates have been recently identified. Heteroresistance (HR) is a phenomenon in which sub-populations of isogenic bacteria exhibit a range of susceptibilities to a particular antibiotic [5,6]. Currently, the mechanisms of CST-HR are not fully understood; however, diverse publications relate this phenotype to traditional CST-resistance mechanisms, which correspond to mutations in the regulatory gene *mgrB* and/or in two-component systems, such as PhoPQ, PmrAB, or PmrC, among others [6]. Specifically, *phoPQ* or *pmrAB* mutations can lead to the constitutive overexpression of the *pmrHFIJKLM*, *pmrC*, or *arnBCADTEF-pmrE* operons involved in phosphoethanolamine (PetN) and 4-amino-4-deoxy-L-arabinose (L-Ara4N) biosynthesis and their transfer to LipA, thereby increasing the positive charges of the LPS and resulting in a decreased affinity to polymyxins [7]. On the other hand, alterations in *mgrB* due to deletion or disruption mediated by IS1-like, IS3-like, and IS5-like elements are the most important CST-HR mechanisms in *K. pneumoniae*. In this sense, *mgrB* is a conserved gene 141 nucleotides in length, which encodes a small transmembrane protein of 47 amino acids that exerts a negative feedback on the PhoPQ and PmrAB systems [8,9]. Specifically, when *mgrB* is disrupted, this inactivated gene is reported to upregulate the PhoPQ or PmrAB systems, thus conferring resistance to CST [9]. Moreover, a novel CST-resistance mechanism has been described in *K. pneumoniae* ST11, ST29, and ST258 lineages, which is related to CrrAB mutations, similar to PhoPQ or PmrAB alterations [10,11]. Interestingly, the CST-HR phenotype may be associated with other no-chromosomal mechanisms, such as capsule-hyperproduction, reducing the interactions of CST with bacterial surfaces, as well as increased expression of RND-type efflux pumps [12,13]. Notably, efflux-pump-mediated CST resistance has not yet been related to CST-HR phenotype expression [6]. CST-HR in *K. pneumoniae* was described previously [14,15,16,17,18] in clinical highly resistant strains, and this phenotype was related to an increase in morbidity and mortality [14], mainly associated with a loss of colistin activity.

Therefore, the identification of CST-HR is a challenging problem for microbiology laboratories, since the usual susceptibility tests fail to detect HR strains. This could lead to the intra-treatment selection of more resistant isolates, recurrent and chronic infections, and, ultimately, increased mortality [14]. There are some published methods to detect HR sub-populations. Among these methods, population analysis profile (PAP) is the “gold standard” technique, as it allows the detection and quantification of resistant sub-populations [19]. Due to the above, the aim of this study was to characterize the CST-HR phenotype among a collection of ESBL-producing *K. pneumoniae* clinical strains collected in Chilean hospitals.

## 2. Materials and Methods

### 2.1. Bacterial Isolates and Antibiotics-Susceptibility Tests

Sixty CST-susceptible and third-generation cephalosporin (3GC)-resistant *K. pneumoniae* isolates collected between 2011 and 2014 from seven Chilean hospitals were included. Bacterial data, such as the type of infection, hospital section, and pathological product, were recorded. Individual patient data were omitted. All isolates were originally identified as *K. pneumoniae* by each hospital laboratory. Antibiotic-susceptibility tests were performed on carbapenems (imipenem (IPM), meropenem (MEM) and ertapenem (ETP)), cephalosporins (ceftriaxone (CRO), cefotaxime (CTX), ceftazidime (CAZ) and cefepime (FEP)), amoxicillin/clavulanic acid (AMC), aminoglycosides (amikacin (AMK) and gentamicin (GEN])), fluoroquinolones (ciprofloxacin (CIP) and levofloxacin (LEV)), tetracycline (TET), and sulfamethoxazole/trimethoprim (SXT). Susceptibility tests were performed by the Kirby–Bauer method, whereas ESBL production was determined by a combined disc test [20]. In all strains positive for ESBL production, β-lactamases genes were screened by conventional PCR [21]. CST susceptibility was determined via the broth microdilution method according to the The European Committee on Antimicrobial Susceptibility Testing—EUCAST guidelines, which considers a minimum-inhibitory concentration (MIC) value of ≤2 μg/mL as CST susceptible [22].

### 2.2. Detection of Colistin-Heteroresistant (CST-HR) Sub-Populations

CST-HR was determined by the PAP technique for all CST-susceptible and ESBL-producing *K. pneumoniae* strains, according to Halaby et al. [15], with the modifications published by Thomas et al. [23]. Briefly, the isolates were inoculated in Müeller–Hinton (MH) broth and incubated under agitation (350 rpm) for 12–18 h at 35 °C ± 2 °C. Then, the grown cultures were standardized to 0.5 of the McFarland standard and diluted from 10^−1^ to 10^−6^ in sterile distilled water. Later, between 10 and 15 micro-drops (2 μL approximately) were inoculated in 6 spots on the surfaces of MH agar plates containing 0, 0.5, 1, 2, 4, 8, and 32 μg/mL of CST. After 48 h of incubation at 35 °C ± 2 °C, the colonies were counted, and the log_10_ CFU/mL was plotted against the CST concentration. The limit of quantification (LOQ) was 2.6 log_10_ CFU/mL. Afterward, we determined the stability of the CST-resistant sub-populations through consecutive sub-cultures of single colonies that appeared on the agar plates supplemented with the highest CST concentration in the PAP experiment. Sub-cultures were performed for up to five generations in MH agar without CST. Then, we determined the CST-MIC of the resulting sub-cultured colonies as described above.

### 2.3. Molecular Strain Typing

To ascertain isogenicity, we determined the clonal relationship between the resulting CST-susceptible and CST-HR sub-populations via ERIC-PCR using an ERIC2 primer (5′-AAGTAAGTGACTGGGGTGAGCG-3′) [24]. The amplified products were separated by 1.5% agarose gel electrophoresis at 60 V for 3 h. A band profile analysis was performed utilizing the GelJ software [25], and the resulting dendrogram was built by the unweighted pair group mean method (UPGMA) using the Dice similarity coefficient and a 1% band position tolerance. The Dice coefficient considered a >90% similarity as indistinguishable strains [26].

### 2.4. Growth Curves

Bacterial growth curves were determined in an MH broth (MHB) that was Mg^2+^/Ca^2+^ cation adjusted. Briefly, overnight cultures of *K. pneumoniae* isolates were diluted in MHB to reach a turbidity equal to 0.5 McFarland. Then, 100 μL of each diluted culture was pipetted into a 96-well plate with 100 μL MHB supplemented with Mg^2+^ and Ca^2+^ at 10 and 25 mg/L, respectively. Following this, OD_650nm_ was measured for 18 h at 37 °C. Each isolate was cultured in triplicate. The growth parameters of K, N_0,_ and g were determined by the GrowthCurve R software, and the Wilcoxon test was used to compare the means between the susceptibly and resistance parameters, with significance set at *p* < 0.05 [27].

### 2.5. Whole-Genome Sequencing (WGS) and in Silico Genome Analyses

Total DNA for whole-genome sequencing (WGS) was extracted using the Wizard^TM^ Genomic DNA Purification kit (Promega, Madison, WI, USA) following the manufacturer’s protocol. DNA concentration and integrity were verified by a spectrophotometer (Take3 BioTek Instruments^TM^, Winooski, Vermont, USA). Sequencing was performed by the Illumina MiSeq platform (2 × 250 bp paired end reads) with libraries prepared by NexteraXT kit (Illumina, San Diego, CA, USA), with coverage of 30×. De novo assembly was carried out using SPAdes assembler version 3.9 (https://cge.cbs.dtu.dk/services/SPAdes/). The assembled genomes were used to screen for antibiotic-resistance determinants with the ResFinder v3.2 tools available at the Center for Genomic Epidemiology server (https://cge.cbs.dtu.dk/services/). In addition, genome annotation was accomplished using the NCBI Prokaryotic Genome Annotation Pipeline (PGAP) web-service (http://www.ncbi.nlm.nih.gov/genome/annotation_prok). Sequence types (STs) were determined for *K. pneumoniae* using bioinformatic web tools (https://cge.cbs.dtu.dk/services/MLST/), and capsular serotypes were identified by the Kaptive web tool [28].

CST-resistance mechanisms were the focus of the analysis of the *mgrB*, *phoPQ*, *pmrAB* genes. Their sequences were analyzed with the local BLAST+ DB using the UGENE software version 1.32.0, including the *K. pneumoniae* MGH78578 (Genbank accession number NC_009648.1) genome as a CST-susceptible reference. The PROVEAN (Protein Variation Effect Analyzer, available at http://provean.jcvi.org/index.php) software was later used to predict whether the amino acid substitutions resulting from missense mutations in *mgrB*, *pmrAB*, and *phoPQ* might affect the functions of these proteins. These genomes have been deposited at the DDBJ/ENA/GenBank under the accession numbers JAAIWP01, JABJWF01, JAAIWO01, JAAIWQ01, JAAIWR01, JABJUO01, JAAFZD01, and JABJUP01.

### 2.6. Transcriptional Analysis by qRT-PCR

Quantitative real-time PCR (qRT-PCR) was used to determine the relative expression of *phoP* (phoP-F 5′-CAG GGA AGC GGA CTA CTA TCT-3′; phoP-R 5′-GCG GCG GAT CAG TGA TAA AAA-3′), *phoQ* (phoQ-F 5′-CCG ACG GTG ACC CTT ATC TAA-3′; phoQ-R 5′-CCA TTG CGT TTC AGC CAT TCC-3′), *pmrD* (pmrD-F 5′-GCA ATC TGG TAT CGC CTT CTA-3′; pmrD-R 5′-CCG GGC AAC AGG ATT ACA-3′), *pmrA* (pmrA-F 5′-AAT CAG CGT CGG CAA TCT-3′; pmrA-R 5′-GAC AGC AGG GCA TAC TCT TTT-3′), *pmrB* (pmrB-F 5′-TGG CGA TGC GAC GTT AAT-3′; pmrB-R 5′-CAT CAG GCC CGC TTT CAA-3′) and *mgrB* (mgrB-F 5′-CTG CCT GTT GCT GTG GAA-3′; mgrB-R 5′-GTG CAA ATG CCG CTG AAA-3′). The primers were designed using the Primer3 design tool.

qRT-PCR reactions were carried out as follows. A culture volume of 5 mL was taken after 18 h incubation at 37 °C (150 rpm) in cation-adjusted MHB with (heteroresistant strains) or without CST (susceptible strain). Next, 2 mL of each culture was centrifuged at 4000 rpm for 15 min. Later, the resulting pellets were used for total RNA extraction with a Trizol^®^ according to the manufacturer’s instructions (Invitrogen, ThermoFisher^TM^, Waltham, MA, USA). qRT-PCR experiments were carried out using a StepOne^TM^ cycler (Applied Biosystems^TM^, Waltham, MA, USA), and a Kapa SYBR Fast qPCR master mix (KapaBiosystems^TM^, Cape Town, South Africa) was used as a signal reporter. For each determination, sterile distilled water was used as a blank. The relative gene expression was measured based on the real-time PCR efficiency (E) and ∆CT from the target and control genes, using *rpoB* (rpoB-rtF 5′-CGCGCAGACCAACGAATATG-3′; rpoB-rtR 5′-CGCCTGAGCGATAACGTAG-3′) as an endogenous control.

## 3. Results

### 3.1. Strain Characteristics and Antimicrobial Susceptibility

Sixty ESBL-producing *K. pneumoniae* determined by a combined disc test and CST-susceptible isolates (CST-MIC_50_ = 1 μg/mL) recovered from seven different Chilean hospitals were included. Of these isolates, 38% (24/60) were collected from medical units and 33% (20/60) from intensive-care units (ICUs). The remaining samples were recovered from ambulatory, surgical, and pediatric units (13% (8/60), 8% (5/60), and 5% (3/60), respectively). Moreover, 52% (31/60) were isolated from urine, 17% (10/60) from blood, 13% (8/60) from sputum, 11% (7/60) from skin and soft tissue, and 7% (4/60) from pleural effusion samples. Antibiotic-susceptibility tests revealed that about 90% were resistant to 3GC (54/60 CRO, 55/60 CTX, 54/60 CAZ), 85% (51/60) were resistant to FEP, 27% to carbapenems (9/60 MEM, 40/60 ETP and all isolates were susceptible to IPM), 98% to fluoroquinolones (59/60 CIP and 58/60 LEV), and 67% to aminoglycosides (28/60 AMK and 52/60 GEN). Moreover, 48% (29/60) of the isolates harbored genes encoding for enzymes of the *bla*_CTX-M-2_ ESBL group, whereas 33% (20/60) contained genes of the *bla*_CTX-M-1_ ESBL group.

### 3.2. Colistin-Heteroresistant Sub-Populations and Drug Resistance

As mentioned above, the population analysis profile (PAP) is the “gold standard” method for the detection of heteroresistant sub-populations [5]. Accordingly, the PAP assay showed the presence of the CST-HR phenotype in 8/60 strains (UCO505, UCO509, UCO511, UCO513, UCO515, UCO517, UCO519, and UCO521) (Table 1). Figure 1a shows the bactericidal curve for the detected sub-populations, evidencing their ability to grow under the highest CST concentration (16 μg/mL). From these results, we determined the frequency of CST-resistant colonies, which oscillated between 10^−5^ and 10^−7^. Interestingly, after five CST-free sub-cultures in MH agar, the resulting CST-MICs_50_ totaled 64 μg/mL in a wide range from 8 to >64 μg/mL, which suggests that the heteroresistant phenotype is stable (Table 2). Furthermore, the growth curves revealed that all strains had the same growth rate constant (K_CST-susceptible_ = 0.02 min^−1^ ± 0,004 versus K_CST-HR_ = 0.02 min^−1^ ± 0.003; *p* = 0.672) and generation time (g_CST-susceptible_ = 32.8 min ± 5.9 versus g_CST-HR_ = 33.6 min ± 4.2; *p* = 0.675), indicating that the heteroresistant phenotype does not affect bacterial fitness (Figure 1b).

The MLST analysis showed that the eight CST-HR *K. pneumoniae* strains belonged to three different lineages: ST11 (UCO511; UCO521), ST25 (UCO509; UCO513; UCO515; UCO519), and ST1161 (UC505; UCO517) (Table 2). Importantly, ST11 is the most disseminated MDR lineage worldwide, whereas ST1161 is apparently endemic to Chile and has not been previously related to CST resistance. On the other hand, ST11 isolates differ in their K-*loci* (KL-15 and KL-81 to UCO521 and UCO511, respectively), whereas ST25 is related to the same and most prevalent capsular serotype (KL-2) (Table 2). For the clonal relationship between the susceptible and CST-HR-derived strains, ERIC-PCR showed that both are closely genetically related, with a Dice’s similarity coefficient up to 90%, confirming that both are isogenic strains (Appendix A, Figure A1).

All CST-HR and ESBL-producing *K. pneumoniae* strains (*n* = 8) exhibited an MDR phenotype and were resistant to several antibiotic groups, including third-generation cephalosporins (7/8 CRO, 7/8 CTX, 8/8 CAZ), FEP (8/8), fluoroquinolones (8/8 CIP and 8/8 LEV), aminoglycosides (5/8 AMK and 6/8 GEN), and SXT (7/8) (Table 2). Interestingly, all were susceptible to IMP, 4/8 to MEM, and 6/8 to ETP. These data are concordant with the in silico analysis, in which we identified the ESBL-encoding genes *bla*_CTX-M-15_ (3/8), *bla*_CTX-M-2_ (3/8)*, bla*_SHV-12_ (1/8), *bla*_SHV-110_ (3/8), and *bla*_OXA-10_ (1/8), which mediate resistance to third-generation cephalosporins and FEP (Table 2). Additionally, we detected non-ESBL enzyme genes, such as *bla*_TEM-1_*, bla*_OXA-2_, and others (Table 2). Moreover, none of the carbapenemase-encoding genes were identified, which suggests that ETP and MEM non-susceptibility may be related to a combination of more than a single mechanism, such as porin loss and ESBL production [29]. In the case of aminoglycoside resistance in CST-HR strains, the in silico analysis revealed the presence of multiple aminoglycosides-modifying enzyme genes, such as N-acetyltransferases (*aac(3)-Ia*, *aac(6’)-Ib3* and *aac(6′)-Ib-cr*), O-nucleotydyltransferases (*ant(3″)-Ia*, *aadA1* and *aadA2*), and O-phosphotransferases (*aph(3′)-Ia*, *aph(3″)-Ib*, *aph(6)-Ib*, and *aph(6′)-Ib*) (Table 2). These results are congruent with the resistant patterns, where a single strain (UCO509) was susceptible to AMK and GEN. Interestingly, the CST-HR UCO517 strain harbored the 16S rRNA methyltransferase *rmtG* gene (Table 2), which is reported to be highly prevalent in South America [30] and was previously reported in a *K. pneumoniae* strain in Chile [30]. The presence of the N-acetyltransferases variant *aac(6′)-Ib-cr* is important since it mediates additional resistance to fluoroquinolones [31]. This gene was present in 6/8 of the CST-HR isolates, which was previously reported in Chile in ESBL-producing *E. coli* and *K. pneumoniae* strains [32]. Accordingly, all CST-HR strains were resistant to CIP and LEV (Table 2); this resistance could be mediated by the multiple resistance determinants detected, including the *aac(6′)-Ib-cr* gene, *oqxAB* genes, and *qnrB1* and *qnrB2* genes. On the other hand, no gyrase mutations associated with quinolone resistance were detected in these strains (Table 2).

### 3.3. Colistin Resistance in Colistin-Heteroresistant K. pneumoniae Strains

To characterize the mechanisms of CST resistance in the CST-HR-derived isogenic strains, the strains were subjected to WGS. The in silico analysis revealed the presence of a mutation in *phoP* in all ST25 CST-HR isolates, corresponding to a T104A amino acid substitution (Table 3). Moreover, this mutation was classified as neutral by the PROVEAN software. For the PhoPQ two-components system, a single isolate (UCO517) had the A351N mutation in PhoQ (Table 3), in which the CST-HR-derived strain showed a 16-fold increase of the CST-MIC. From the PROVEAN analysis, this substitution was predicted to have a deleterious effect on the protein function. Additionally, UCO519 presented a P95L amino acid substitution in the PmrB protein (a 16-fold CST-MIC increase), whereas A256G was detected in both UCO511 and UCO521, in which a 32-fold CST-MIC increase was detected (Table 3). This mutation was also considered to be deleterious by PROVEAN. Furthermore, the UCO511 and UCO521 strains belong to the ST11 lineage; however, they differ in their K-*loci* (KL-81 and KL-15m, respectively) (Table 2). Moreover, the ERIC-PCR analysis showed a <90% similarity; consequently, these isolates were considered genetically unrelated (Appendix A, Figure A1). Four CST-HR-derived isolates showed alterations on the *mgrB* gene, which is known to mediate resistance to CST (Table 3). Specifically, a single nucleotide modification was identified in UCO521, which corresponds to the deletion of 27-bp in the gene, whereas the other three CST-HR-derived strains (UCO505, UCO511, and UCO513) displayed a disruption of the *mgrB* gene mediated by the insertion sequences IS5-like and IS1-like (Table 3). Additionally, no amino acid substitutions were identified in the PmrA and PmrD proteins in these isolates, and *mcr*-like genes were also absent.

### 3.4. qRT-PCR Analysis

The expression levels of CST-related resistance genes were analyzed for CST-susceptible and CST-HR-derived strains to evaluate the effect of *pmrB*, *phoP*, and *mgrB* mutations (Figure 2). Accordingly, we observed a decrease of the relative *mgrB* expression in heteroresistant UCO505, UCO511, and UCO519, which are related to IS element insertion but not significant modifications of expression in UCO521, which presented a premature stop codon (Figure 2).

On the other hand, all isolates, except for UCO509, displayed a significantly increased relative expression of the *phoP* gene in their CST-HR-derived isogenic strains (Figure 2), which is related to LPS modification mediated by the *pmrFHIJKLM* operon [9]. The *phoP* gene has been associated with increased expression of *prmD* [9], which was observed in CST-HR-derived UCO505 and UCO517 isolates. Furthermore, both UCO505 and UCO519 showed a significantly increased relative expression of *phoQ* (Figure 2), a sensor kinase, which activates *phoP* via phosphorylation [13]. Moreover, we observed an increase in the expression of *pmrB*, especially in CST-HR-derived UC511, UCO519, and UCO521 strains (Figure 2), which may be due to amino acid substitution detected by WGS (Table 2). In addition, only the CST-HR UCO521 strain evidenced a significant increase in *pmrA* expression (Figure 2), which was reported to induce LipA modifications in CST-resistant isolates [31].

## 4. Discussion

*K. pneumoniae* is an important human pathogen, which is classified as a common opportunistic hospital-associated bacteria involved in multiple infections, including urinary tract infections, cystitis, pneumonia, and surgical wound infections [33]. During the last decade, there has been a notable increase in the reports of highly resistant *K. pneumoniae* isolates worldwide [34]. In our study, we included 60 *K. pneumoniae* MDR-strains that displayed resistance to multiple antibiotics. In highly resistant *K. pneumoniae* isolates, one of the most important resistance mechanisms corresponds to the presence of ESBLs enzymes, which have a global prevalence of 20% to 70% [35]. Specifically, the CTX-M ESBL family is the most prevalent family globally, in which the variant CTX-M-2 group is the predominant ESBL in *K. pneumoniae* isolates from South America [32,36], which is concordant with our findings.

Even though the HR phenotype was reported 50 years ago, it has been extensively studied only during the last five years in Enterobacterales members [6]. In general terms, antibiotic HR is a process in which sub-populations of isogenic bacteria display different ranges of susceptibilities to a certain drug [18]. Even though the mechanisms involved in HR are unclear, it has been proposed that this phenomenon could be promoted by the intra-treatment usage of antibiotics [14]. Even though antibiotic HR affects clinical outcomes [14], its study is limited mainly because it cannot be easily explored in standard clinical laboratories [18]. Because the PAP test, considered to be the “gold standard,” is time-consuming and costly, other methods have been proposed, such as combined microscopy and microfluidic methods, simplified PAP and differential antibiotic-susceptibility tests. However, Andersson et al. proposed a new technique combining susceptibility tests and different inoculum sizes, which has been shown to improve the detection of HR bacteria [6]. Due to the above, few epidemiological data exist on HR prevalence [14]. Thus, our study provides more information on this phenomenon in clinically relevant pathogens. Specifically, our study is the first CST-HR *K. pneumoniae* description in South America. Although the global frequency of this phenotype is still unknown, our study identified 13% CST-HR in a collection of 60 non-repetitive *K. pneumoniae* samples. Previously, Cheong et al. investigated the presence of the CST-HR phenotype in South Korea in a collection of 231 CST-susceptible isolates recovered from bloodstream infections and found a frequency of 1.3% [18], which is significantly lower compared to the 72% frequency reported by Meletis G et al. [17]. Nicoloff et al. identified 27% of clinical isolates that show a CST-HR phenotype in four different bacterial species, including *K. pneumoniae* [37]. These inconsistencies may be due to the difficulty in detecting CST-HR, partly because of the different detection methodologies employed [6]. In our study, we utilized the PAP technique, which is considered the “gold standard” method for identifying heteroresistant sub-populations, as other methods, such as E-tests, fail to detect CST-HR sub-populations because the proportion of HR cells is normally low [17,38]. Poudyal et al. reported a CST-HR proportion in *K. pneumoniae* between 6.03 × 10^−9^ and 1.29 × 10^−5^ [16], whereas an HR proportion between 10^−3^ and 10^−6^ was reported in carbapenem-resistant and CST-HR *K. pneumoniae* [17]. Our results are concordant with those previously reported since we determined a low CST-HR proportion varying between 10^−5^ and 10^−7^ cells. On the other hand, the observed HR phenotype is different to the persistence phenomenon, which mediates antibiotic tolerance at the associated cost of bacterial growth [14,39]. Remarkably, our results showed no differences in the growth rates between parental (CST-susceptible) and CST-resistant sub-populations, which suggests that the observed CST-HR phenotype does not correspond to CST-tolerance, as reported for CST-HR *Acinetobacter baumannii* [40]. Furthermore, our results confirm that the CST-HR phenotype is stable since we observed an increase in CST-MIC values by at least two-fold after more than five generations without antibiotic selective pressure [6,39]. As mentioned in the results section, we were able to generate CST-resistance isogenic strains on eight ESBL-producing *K. pneumoniae* isolates. To determine the genetic mechanisms involved in this process, we investigated the two-component systems of PmrAB and PhoPQ, in addition to the global regulator MgrB, which corresponds to the main chromosomal mechanisms involved in CST-resistance in this species [41]. Mutations in the *pmrB* gene have been widely described in *K. pneumoniae* isolates that are resistant or have reduced susceptibility to CST [42]. Pitt et al. determined that a punctual mutation in *pmrB* of an extensively drug-resistant (XDR) *K. pneumoniae* ST258 isolate can yield amino acid substitution in P95L, which is consequently related to a significant increase of the MICs to CST, demonstrating that a single mutation can confer resistance to this drug [43]. Moreover, Jayol et al. described a single W157P amino acid substitution in the *pmrB* of six CST-resistant isolates. The authors suggested that this substitution could have an impact on the dimerization process of PmrB, which might induce the constitutive activation of PmrA, thereby producing the overexpression of *pmrC*, and ultimately leading to CST resistance [44]. The role of *pmrAB* mutations in the CST-HR phenotype has been demonstrated in *A. baumannii* and *Pseudomonas aeruginosa*, in which a single amino acid substitution in PmrB was associated with HR [45,46]. Remarkably, mutations in *pmrB* may arise after low-dosage CST exposure, which could produce a stable CST-resistant phenotype without a fitness cost [47]. Our results showed that the UCO519 isolate (ST25) possessed a P95L amino acid substitution in PmrB, whereas an A256G amino acid substitution in UCO511 and UCO521 (both ST11) was identified without an evident fitness cost. Cheng et al. demonstrated that the same substitution was not sufficient to alter the MICs to CST; secondary factors contributing to CST resistance are likely necessary [47].

PhoP is a member of the PhoPQ two-component system involved in CST resistance since it activates the *pmrHFIJKLM* operon, thereby conferring resistance to this drug [31]. Our findings revealed an amino acid substitution in position 104 in PhoP, which was detected in the CST-HR isolates belonging to ST25. To the authors’ best knowledge, this mutation has not been previously described in CST-resistant *K. pneumoniae* isolates. Thus, further studies are required to determine its role in CST resistance. For the HR isolate that belongs to ST1161 (UCO517), we identified an A351D amino acid substitution in PhoQ. Jayol et al. reported a CST-HR mechanism in *K. pneumoniae* that involves a single substitution in PhoQ (A191Y), which generates a structural modification in this protein, thereby producing the CST-HR phenotype [48]. Furthermore, Halaby et al. identified an A21S substitution in PhoQ in a CST-HR and ESBL-producing *K. pneumoniae* isolate, which produced an eight-fold increase of CST-MIC from 2 to 16 μg/mL, demonstrating that this amino acid substitution plays an important role in the CST-HR phenotype [15]. Moreover, Kim et al. described multiple amino acid substitutions in PhoPQ, PmrB, and MgrB that emerged after exposure to CST, such as during the development of CST-HR [49]. In addition to the previously described mechanisms, alterations in the MgrB global regulator have been identified as the main chromosomal mechanism of CST resistance in *K. pneumoniae* [49,50]. Specifically, the *mgrB* gene encodes a negative-feedback regulator of the PhoPQ two-component system, whose inactivation is associated with the upregulation of the PhoPQ system, resulting in modifications of the LPS and, consequently, resistance to CST [41]. In this respect, it has been observed that *mgrB* inactivation by IS-like elements reduces *mgrB* mRNA levels, which is associated with CST resistance [49]. Our results showed that three CST-HR isolates possessed IS5-like (UCO505) and IS1-like (UCO511 and UCO513) transposons—insertion sequences that were previously described in CST-resistant isolates [49]. These results are congruent with the *mgrB* expressions levels determined by RT-qPCR.

## 5. Conclusions

CST-HR represents a threat to treating serious infections caused by relevant nosocomial pathogens, since CST is a “last-resort” alternative to control highly resistant infections. This is the first report on eight CST-HR ESBL-producing *K. pneumoniae* isolates in South America. Although the proportion of CST-HR detected in our study was low, it remains of special concern since CST-HR was not previously identified in a clinical setting in Chile. Here, the resulting CST-resistant isolates were stable, displaying high levels of resistance to CST. The genetic mechanisms involved in this resistance included mutations in PhoPQ and PmrAB, some of which had not been previously identified. The disruption of MgrB was identified, while plasmid-contained *mcr*-like genes were not found. Since this phenotype has been poorly researched, further studies are necessary to understand its mechanisms and epidemiology, since the rise of CST resistance will leave limited therapeutic options to treat this important pathogen.

## Figures and Tables

**Figure 1 microorganisms-08-01279-f001:**
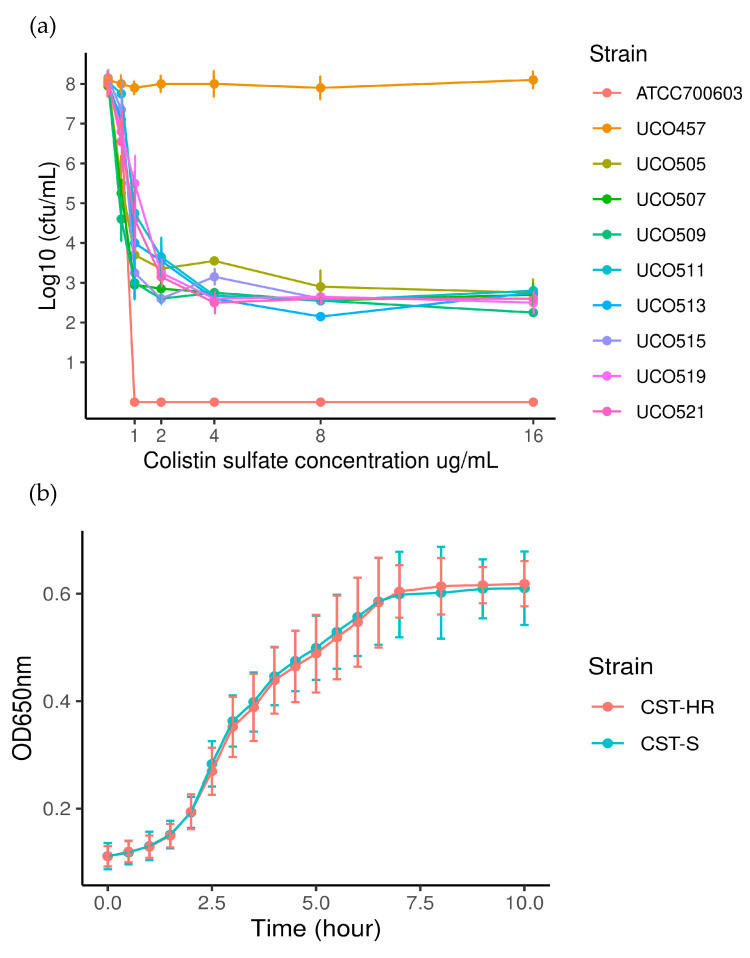
(**a**). Population analysis profiles (PAPs) for CST-HR strains. ATCC 700603 (*Klebsiella quasipneumoniae subsp. similipneumoniae*) was used as a negative control (CST-susceptible) and *E. coli* UCO457 as a positive control (CST-resistant *mcr-1* positive); (**b**). bacterial growth curves for CST-susceptible (CST-S) and heteroresistant (CST-HR) strains.

**Figure 2 microorganisms-08-01279-f002:**
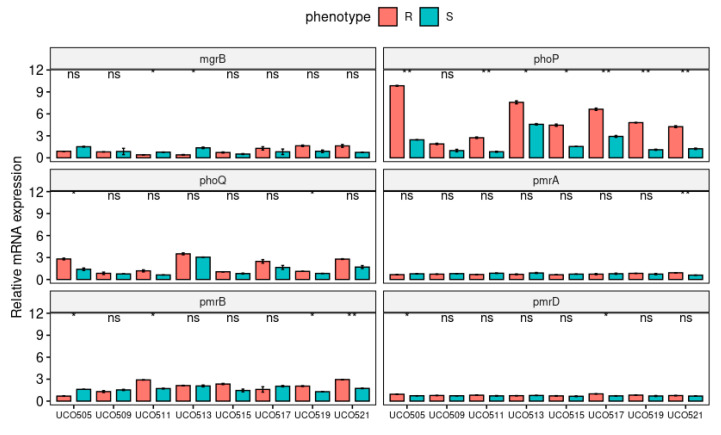
Relative expression (RT-qPCR) of the CST-resistance genes *mgrB*, *phoPQ*, and *pmrABD* in CST-susceptible (S) and CST-HR-derived (R) *K. pneumoniae* strains. * *p* < 0.05; ** *p* < 0.01, ns: not significant.

**Table 1 microorganisms-08-01279-t001:** Data of the Colistin-heteroresistant (CST-HR) extended-spectrum β-lactamase (ESBL)-producing *K. pneumoniae* isolates collected in Chilean hospitals.

Strain	Year of Isolation	Location	Unit	Sample	GenBank Accession Number
UCO511	2012	Concepción	ICU	Urine	JAAIWP01
UCO521	2013	Santiago	ICU	Catheter	JABJWF01
UCO509	2012	Los Angeles	MED	Blood	JAAIWO01
UCO513	2012	Valparaíso	ICU	Blood	JAAIWQ01
UCO515	2013	Santiago	ICU	Urine	JAAIWR01
UCO519	2013	Santiago	SUR	Blood	JABJUO01
UCO505	2012	Puerto Montt	MED	Bronquial	JAAFZD01
UCO517	2012	Santiago	MED	Urine	JABJUP01

ICU: Intensive-care unit; MED: medicine; SUR: surgery.

**Table 2 microorganisms-08-01279-t002:** Main characteristics of CST-HR *K. pneumoniae* strains (*n* = 8). The table includes antibiotic-susceptibility patterns, clonal lineages, and antibiotic-resistance genes.

			CST-MIC (g/mL)			Antibiotic Resistance Genes and Resistance Patterns		
Strain	ST	K-locus	S	HR	CST-HR Frquency (10^−6^)	β-lactamases (*bla*)	Aminoglycosides Resistance Genes	Resistance Pattern
UCO 511	11	81	2	64	5.30	*CTX-M-2; OXA-2; SHV-182; TEM1*	*aac(3)-IIa; aac(6′)-Ib3; aadA2; aph(3″)-Ib; aph(6)-Id*	ERT; CPD; CRO; CTX; CAZ; FEP; AMC; GEN; AMK; LEV; CIP
UCO 521	11	15	2	64	6.70	*CTX-M-15; OXA-1; SHV-182*	*aac(3)-IIa; aac(6′)-Ib-cr; aadA2; aph(3′)-Ia*	CPD; CRO; CTX; CAZ; FEP; GEN; LEV; CIP
UCO 509	25	2	1	64	3.30	*SHV-110; SHV-81; TEM-1B*	*aph(3″)-Ib; aph(6)-Id*	CAZ; FEP; AMC; LEV; CIP
UCO 513	25	2	0.5	32	5.60	*CTX-M-2; OXA-10; SHV-12; TEM1B*	*aadA1; aph(3″)-Ib; aph(6)-Id*	ERT; CPD; CRO; CTX; CAZ; FEP; AMC; GEN;LEV; CIP
UCO 515	25	2	2	>64	2.80	*CTX-M-15; OXA-1; OXA-10; SHV-110; TEM-1B*	*aac(3)-IIa; aac(6´)-Ib-cr; aadA1; aph(3″)-Ib; aph(6)-Id*	ERT; CPD; CRO; CTX; CAZ; FEP; AMC; GEN; AMK; LEV; CIP
UCO 519	25	2	2	32	3.30	*CTX-M-15; OXA-1; OXA-10; SHV-110; SHV-81, TEM-1B*	*aac(3)-IIa; aac(6´)-Ib-cr; aadA1; aph(3″)-Ib; aph(6)-Id*	ERT; CPD; CRO; CTX; CAZ; FEP; AMC; GEN; AMK; LEV; CIP
UCO 505	1161	19	2	64	0.32	*CTX-M-2; OXA-2; TEM-1B*	*aac(3)-IIa; aac(6′)Ib3; aac(6′)-Ib-cr*	ERT;CPD;CRO;CTX;CAZ;FEP;AMC;GEN;AMK;LEV; CIP
UCO 517	1161	19	0.5	8	0.14	*OXA-10; OXA-9; SHV-187; TEM-1A*	*aac(6′)-Ib; aadA1; rmtG*	ERT; CPD; CRO; CTX; CAZ; FEP; AMC; GEN; AMK; LEV; CIP

ST: Sequence type, HR: heteroresistant, S: susceptible, HR: heteroresitance strain, imipenem (IPM), meropenem (MEM), ertapenem (ETP), ceftriaxone (CRO), cefotaxime (CTX), ceftazidime (CAZ), cefepime (FEP), amoxicillin/clavulanic (AMC), amikacin (AMK), gentamicin (GEN), ciprofloxacin (CIP), levofloxacin (LEV), tetracycline (TET), sulfamethoxazole/trimethoprim (SXT).

**Table 3 microorganisms-08-01279-t003:** Modification of CST-resistance genes in CST-HR-derived *K. pneumoniae* isolates.

		MIC (ug/mL)		PmrB		PhoP	PhoQ	MgrB	
Strains	ST	S	HR	95	256	104	351	39	IS
UCO505	1161	2	64						IS5-like
UCO517	1161	0.5	8				Ala→Asp		
UCO509	25	1	64			Thr→Ala			
UCO513	25	0.5	32			Thr→Ala			IS1-Like
UCO515	25	2	>64			Thr→Ala			
UCO519	25	2	32	Pro→Leu		Thr→Ala			
UCO511	11	2	64		Arg→Gly				IS1-Like
UCO521	11	2	64		Arg→Gly			Cys X	

ST: Sequence type; S: susceptible; HR: heteroresistant; MIC: minimum-inhibitory concentration; →: substitution; IS: insertion sequence, Cysx. stop codon.

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
