# Peer review of "Colistin Heteroresistance among Extended Spectrum β-lactamases-Producing Klebsiella pneumoniae"

_microorganisms, 2020, doi:10.3390/microorganisms8091279_

Round 1
Reviewer 1 Report
The authors provide their results of colistin-heteroresistances among K. pneumoniae isolates derived from seven Chilean hospitals. After a screening for heteroresistant subpopulations via PAP resulting in 8 of 60 HR-strains, these were further characterized geno- and phenotypically.
Despite the topic is of interest, some aspects need to be addressed:
Major points:
- To ascertain isogenicity of susceptible strains and HR-subpopulations an ERIC-PCR was performed and a resulting Dendrogramm is presented. Additionally WGS is used for detection of MLST ST and resistance mechanisms. One cannot understand, why WGS data was not used to ascertain clonal relationship, e.g. with the help of cgMLST algorithm. Please provide this analysis based on at least the core genome (perhaps whole genome) data.
- Please evaluate your findings regarding the clinical point of view within the discussion section. What are your suggestions for further screening measures, infection prevention activities etc.?
Minor points:
- English language should be edited.
- Style (font size and type) varies frequently within the manuscript. Please adapt.
Author Response
Reviewer #1
The authors provide their results of colistin-heteroresistances among K. pneumoniae isolates derived from seven Chilean hospitals. After a screening for heteroresistant subpopulations via PAP resulting in 8 of 60 HR-strains, these were further characterized geno- and phenotypically.
Despite the topic is of interest, some aspects need to be addressed:
Major points:
- To ascertain isogenicity of susceptible strains and HR-subpopulations an ERIC-PCR was performed and a resulting dendrogram is presented. Additionally WGS is used for detection of MLST ST and resistance mechanisms. One cannot understand, why WGS data was not used to ascertain clonal relationship, e.g. with the help of cgMLST algorithm. Please provide this analysis based on at least the core genome (perhaps whole genome) data.
Dear reviewer, thank you very much for your comments. The main aim of ERIC-PCR was to compare between parental isolates (colistin-susceptible) and their colistin-heteroresistant progeny. Accordingly, we did not sequence the whole-genome of the parental isolates, and we did sequence only the colistin-heteroresistant progeny genomes. The idea of ERIC-PCR was to determine that colistin-heteroresistant progeny isolates effectively derived from colistin-susceptible parental isolates. Unfortunately, due to resources limitations, we did not sequence parental susceptible isolates. Colistin-heteroresistant progeny genomes were compared against K. pneumoniae MGH78578 genome. (Genbank accession number NC_009648.1).
- Please evaluate your findings regarding the clinical point of view within the discussion section. What are your suggestions for further screening measures, infection prevention activities etc.?
Dear reviewer, thank you very much for your comment. We added a paragraph about the impact of heteroresistance and the need for implement methods to measure its impact in public health (Lines )
Minor points:
- English language should be edited.
We submitted the manuscript to the English editing service of MDPI.
- Style (font size and type) varies frequently within the manuscript. Please adapt.
These problems were solved.
Reviewer 2 Report
Morales-Leon et al. investigated 8 Klebsiella pneumoniae isolates with colistin heteroresistance, which were part of an isolate collection from Chile that contained 60 colistin-susceptible K. pneumoniae isolates. The authors used Population Analysis Profiles to screen the isolate collection, and then sequencing and growth curves were used to build upon preliminary drug susceptibilities. Overall, I believe the study is interesting and adds meaningfully to the literature corresponding to polymyxin resistance; however, I have several comments/suggestions that authors should address before the manuscript is published.
Although I commend the authors for a relatively well-written manuscript, there are numerous grammatical errors throughout the paper that are somewhat distracting. I recommend having a native English speaker proofread the document to improve readability.
- Lines 164 – 165: 52% of the isolates were collected from urine, and 17% of the isolates were collected from blood. Can the authors disclose where the other isolates were collected (sputum versus wound debridement, etc.)?
- Lines 165 – 166: I found this sentence very confusing. The authors claim that 73% of the isolates were resistant to third generation cephalosporins, but then go on to state that 54/60 were resistant to ceftriaxone, 55/50 were resistant to cefotaxime, and 54/60 were resistant to ceftazidime. Based on the above fractions, weren’t at least 90% of the isolates resistant to third generation cephalosporins? Also, I assume that 55/50 is a typo and there were actually 55/60 isolates that were resistant to cefotaxime. Moreover, lines 74 – 75 state that “sixty CST-susceptible and third generation cephalosporin-resistant” isolates were used in the study, but it appears that there were isolates that were not resistant to third generation cephalosporins.
- Lines 166 – 167: I was surprised that 40 of the isolates were resistant to ertapenem but none of the isolates were resistant to imipenem. The authors later comment in lines 209 – 210 that carbapenemase enzymes were not detected after whole genome sequencing the heteroresistant isolates. Can the authors provide a sense of how often the aforementioned clinical scenario is encounted in Chile, or perhaps provide a more specific mechanistic hypothesis as to which ESBL enzymes and porin deletions confer resistance to ertapenem but not imipenem?
- Table 1 – The source of some of the samples is listed as “Orine,” but presumably it should read “Urine”
- Figure 1 – I think the authors can make the graph easier to interpret if the X-axis ticks go by increments of 2 or 4 mcg/mL and the final value on the X-axis is either 16 mg/L or a value above 16 mg/L. Likewise, the Y-axis can probably be labeled with consecutive log cfu/ml of 1, 2, 3, 4, 5, 6, 7, 8, and 9. Lastly, the Y-axis label reads “(ufc/ml)” instead of (cfu/ml)
- Lines 178 – 179 state that subculturing the heteroresistant isolates 5 times on drug-free media resulted in a colistin MIC of 32 mcg/ml, but in Table 2 it appears the colistin MICs ranged from 8 to > 64 mcg/ml. Also, line 302 states that the organisms were subcultured 50 times, but I believe the authors intended to write that 5 generations or organisms were used.
- The authors discuss the fitness of the colistin heteroresistant isolates in lines 182 – 183 and lines 317 – 319. Is there anything in the literature that discusses if any of the mutations detected by the authors in the heteroresistant isolates impacts the virulence of pneumoniae or other Enterobacteriacea? The trade of between polymyxin resistance and virulence is well established for Acinetobacter baumannii.
- Figure 2 – Everything in the figure is well presented except that the strain numbers are difficult to read. Can the authors perhaps label the bottom of the figure as “UCO Strain Number” or something of that nature and then report the strain numbers in a larger front?
Author Response
Morales-Leon et al. investigated 8 Klebsiella pneumoniae isolates with colistin heteroresistance, which were part of an isolate collection from Chile that contained 60 colistin-susceptible K. pneumoniae isolates. The authors used Population Analysis Profiles to screen the isolate collection, and then sequencing and growth curves were used to build upon preliminary drug susceptibilities. Overall, I believe the study is interesting and adds meaningfully to the literature corresponding to polymyxin resistance; however, I have several comments/suggestions that authors should address before the manuscript is published.
Although I commend the authors for a relatively well-written manuscript, there are numerous grammatical errors throughout the paper that are somewhat distracting. I recommend having a native English speaker proofread the document to improve readability.
- Lines 164 – 165: 52% of the isolates were collected from urine, and 17% of the isolates were collected from blood. Can the authors disclose where the other isolates were collected (sputum versus wound debridement, etc.)? ANSWER: the data were completed with: 13%(8/60) from sputum, 12% (7/60) skin and soft tissue and 7% (4/60) from pleural effusion samples.
- Lines 165 – 166: I found this sentence very confusing. The authors claim that 73% of the isolates were resistant to third generation cephalosporins, but then go on to state that 54/60 were resistant to ceftriaxone, 55/50 were resistant to cefotaxime, and 54/60 were resistant to ceftazidime. Based on the above fractions, weren’t at least 90% of the isolates resistant to third generation cephalosporins? Also, I assume that 55/50 is a typo and there were actually 55/60 isolates that were resistant to cefotaxime. Moreover, lines 74 – 75 state that “sixty CST-susceptible and third generation cephalosporin-resistant” isolates were used in the study, but it appears that there were isolates that were not resistant to third generation cephalosporins. ANSWER: We selected ESBL strains by combined disc test. we consider a positivecombined disc test to be ESBL production, but when susceptibility was determined, not all strains were resistant to all cephalosporins. It necessary considered that combined disc test has about 96% of sensitivity (Drieux L et al. Clin Microb and Infect 2008;14:supp1:90-103) and the Kirby-bauer method can't always identify C3G resistance, especially when it's near the breaking point. In our case, about 90% was cephalosporins resistance by diffusion disc method. The “73%” and “55/50 cefotaxime resistans strain” was writer error. This was corrected in the text.
- Lines 166 – 167: I was surprised that 40 of the isolates were resistant to ertapenem but none of the isolates were resistant to imipenem. The authors later comment in lines 209 – 210 that carbapenemase enzymes were not detected after whole genome sequencing the heteroresistant isolates. Can the authors provide a sense of how often the aforementioned clinical scenario is encounted in Chile, or perhaps provide a more specific mechanistic hypothesis as to which ESBL enzymes and porin deletions confer resistance to ertapenem but not imipenem?.
ANSWER: The alterations or losses of major non-specific porins is and important antimicrobial resistant mechanism, specially in pneumoniae,how have two main porin (OmpK35 and OmpK36). This mechanism describe in ESBL-producing k. pneumoniae strains, are related with decrease susceptibility in many antibiotic, particularly ertapenem. The loss of susceptibility due by this mechanism is more marked for ertapenem than for imipenem or meropenem, this is probably due by ertapenem, has a larger molecular weight compared to the other carbapenems. https://doi.org/10.1093/jac/dkp029Iin this sense, carbapenem resistance in Chilean isolates is mainly due to a combination of porin loss/alteration and β-lactamase activity, due by the ESBLs and AmpC enzymes have a residual capacity to hydrolyse carbapenems. https://doi.org/10.1099/jmm.0.045799-0. In Chile, the prevalence of blaKPC and other carbapenemase remanins low. (reference)
- Table 1 – The source of some of the samples is listed as “Orine,” but presumably it should read “Urine”.
ANSWER: This mistake was corrected.
- Figure 1 – I think the authors can make the graph easier to interpret if the X-axis ticks go by increments of 2 or 4 mcg/mL and the final value on the X-axis is either 16 mg/L or a value above 16 mg/L. Likewise, the Y-axis can probably be labeled with consecutive log cfu/ml of 1, 2, 3, 4, 5, 6, 7, 8, and 9. Lastly, the Y-axis label reads “(ufc/ml)” instead of (cfu/ml). ANSWER: The graph was made using the revisor recommendation.
- Lines 178 – 179 state that subculturing the heteroresistant isolates 5 times on drug-free media resulted in a colistin MIC of 32 mcg/ml, but in Table 2 it appears the colistin MICs ranged from 8 to > 64 mcg/ml. Also, line 302 states that the organisms were subcultured 50 times, but I believe the authors intended to write that 5 generations or organisms were used.
ANSWER: After subculturing the HR isolates in free-CST media, we determined the MIC by microdilution method, and the MIC50 was 64 ug/mL, in a wide range from 8 to >64ug/ml. The MIC of 32ug/ml its was typing error. Second, you're right, they were sub-cultivated for 5 generation and this was explain in line 100. 50 times it was a misconception. this was corrected in the text.
- The authors discuss the fitness of the colistin heteroresistant isolates in lines 182 – 183 and lines 317 – 319. Is there anything in the literature that discusses if any of the mutations detected by the authors in the heteroresistant isolates impacts the virulence of pneumoniaeor other Enterobacteriacea? The trade of between polymyxin resistance and virulence is well established for Acinetobacter baumannii. ANSWER:In our case, these mutations was not previously described and there is no related information. In this regard, at K. pneumoniae colistin resistant and his impact in virulence, the results are controversial. For example, Kidd T et al, described a relation between polymyxin, specially in mgrB mutation and enhances virulence in pneumoniae due by attenuating early host defense response activation. This have a important implications in virulence activity of K. pneumoniae. https://www.ncbi.nlm.nih.gov/pmc/articles/PMC5376759/. But, for the other hand, Arena F. et al, describe by G. mellonella virulence model, that K. pneumoniae ST258 colistin resistan due by mgrB mutation not impact in virulence profile. Finally, Myung-Jin Choi and Kwan Soo Ko, study the effect on the hypervirulent profile in K. pneumonie ST23 due by colistin-resistance in xxxx and xxx. This mutation could be related with a reduce hipervyrulent profile due by a decrese formation of capsular polysaccharides (CPS) and reduced expression of hypervirulent genes (rmpA/A2). https://www.ncbi.nlm.nih.gov/pmc/articles/PMC4604379/
- Figure 2 – Everything in the figure is well presented except that the strain numbers are difficult to read. Can the authors perhaps label the bottom of the figure as “UCO Strain Number” or something of that nature and then report the strain numbers in a larger front? Answer: For a better reading of graph, the figure 2 was reordered using a 2x3 matrix.
Round 2
Reviewer 1 Report
The manuscript has improved substantially and is suitable for publication now.